Fuzzy inference rule based task offloading model (FI-RBTOM) for edge computing

Ibrahim Kashif 1
Sajid Ahthasham 2 ahthasham.sajid@riphah.edu.pk
http://orcid.org/0000-0002-0864-6215 Ullah Ihsan 1
Ullah Khan Inam 3
http://orcid.org/0000-0003-3777-765X Kaushik Keshav 4
Askar S S. 5
Abouhawwash Mohamed 6 7
1 Department of Computer Science and Information Technology, University of Balochistan , Quetta, Balochistan , Pakistan
2 Department of Cyber Security and Data Science, Riphah Institute of Systems Engineering, Riphah International University , Islamabad , Pakistan
3 Department of Computer Science, National University of Technology , Islamabad , Pakistan
4 Amity School of Engineering and Technology, Amity Univeristy, Punjab , Mohali , India
5 Department of Statistics and Operations Research, College of Science, King Saud University , Riyadh , Saudi Arabia
6 Department of Computational Mathematics, Science and Engineering (CMSE), College of Engineering, Michigan State University , East Lansing , United States
7 Department of Mathematics, Faculty of Science, Mansoura University , Mansoura , Egypt
D’Agostino Daniele
Electronic publication date: 2025 Mar 28
Publication date: 2025
Volume: 11
Electronic Location ID: e2657
Received 2024 Jul 26; Accepted 2024 Dec 23
Copyright: © 2025 Ibrahim et al.
Copyright year: 2025
Copyright holder: Ibrahim et al.
License: This is an open access article distributed under the terms of the Creative Commons Attribution License, which permits unrestricted use, distribution, reproduction and adaptation in any medium and for any purpose provided that it is properly attributed. For attribution, the original author(s), title, publication source (PeerJ Computer Science) and either DOI or URL of the article must be cited.
License URL: https://creativecommons.org/licenses/by/4.0/

Keywords: Edge computing, Cloud computing, Fuzzy logic, Local, FI-RBTOM

Funding: King Saud University, Riyadh, Saudi Arabia RSP2025R167 This work was supported by King Saud University, Riyadh, Saudi Arabia, through the Researchers Supporting Project RSP2025R167. The funders had no role in study design, data collection and analysis, decision to publish, or preparation of the manuscript.

==============================
The key objective of edge computing is to reduce delays and provide consumers with high-quality services. However, there are certain challenges, such as high user mobility and the dynamic environments created by IoT devices. Additionally, the limitations of constrained device resources impede effective task completion. The challenge of task offloading plays a crucial role as one of the key challenges for edge computing, which is addressed in this research. An efficient rule-based task-offloading model (FI-RBTOM) is proposed in this context. The key decision of the proposed model is to choose either the task to be offloaded over an edge server or the cloud server or it can be processed over a local node. The four important input parameters are bandwidth, CPU utilization, task length, and task size. The proposed (FI-RBTOM), simulation is carried out using MATLAB (fuzzy logic) tool with 75% training and 25% testing with an overall error rate of 0.39875 is achieved.

Introduction

A well-recognized soft computing science technique called fuzzy logic incorporates organized human knowledge into algorithms to build feasible solutions. It is a logical framework that gives an approximate, rather than an accurate, model intended for human interpretation styles. Fuzzy logic handles challenges of fuzzy thinking, ambiguity, estimates, unpredictability, qualitative mess, or partial accuracy. The extension of fuzzy logic soft computing is fuzzy “if-then-else” rule-based modeling. Fuzzy rule-based modeling is executed without a feedback process, although fuzzy logic systems are usually constructed with a feedback process (Shruti & Chandra Deka, 2020). Nowadays, edge computing is a prominent field of research (Mouradian et al., 2017). It puts end users or sensors closer to cloud services. For applications used by end users, edge computing has become a new computing paradigm with low latency and high bandwidth (Keyan et al., 2020). With cloud computing, users can move their data from their computer to the cloud via the Internet and make highly effective decisions about processing and central data storage. It is an extremely powerful platform for network services that utilize multiple technologies, such as virtualization, distributed computing, load balancing, parallel computing, and network storage. Nevertheless, in modern times, as the Internet of Things grows more and more integrated into people’s daily lives, an increasing number of devices are connected to it, which generates an enormous quantity of data (Keyan et al., 2020). Consequently, the cloud-computing model has significant flaws regarding load, real-time transmission bandwidth, energy consumption, and the protection of data security and privacy (Li, Tao & Chen, 2020).

Mobile edge computing (MEC) networks depend on two primary components: task offloading and efficient resource allocation. According to Zhou et al. (2017), previous research has categorized these components into three main categories: binary or full offloading (where the task cannot be divided during processing), collaborative task offloading (where the task interacts with the cloud), and partial offloading (where a task is divided into multiple parts for simultaneous offloading or local processing).

To decrease latency and the rate at which computing tasks fail to describe, an accurate fuzzy logic decision-based task offloading method has been suggested to enable the use of Internet of Things applications. A mobile edge orchestrator supported by a fuzzy foundation is in charge of managing end-user applications (Salmani et al., 2007).

To boost the efficiency of managing computational resources, the authors of this article suggested a fuzzy-based mobile edge orchestrator method for flexible computation offloading for IoT applications. By network conditions, computational capacity, and task specifications, a fuzzy-based MEC enables edge workload orchestration actions to decide whether to transfer the mobile user’s tasks to nearby edge, local edge, or cloud servers as described by Li, Tao & Chen (2020).

Chen & Xu (2019) tackles the problem of resource management and offers an intelligent architecture that makes use of software defined networking (SDN) and fuzzy logic (FL) techniques to manage networking, cloud-fog-edge computing, and storage resources effectively and efficiently in vehicular ad hoc networks (VANETs). The proposed SDN-VANETs approach uses fuzzy logic to manage resources in real time while handling ambiguity and inaccuracy. This article presents a robust decentralized task-unloading approach that considers numerous key metrics, including relative velocity, available computing resources, link reliability, mobility, and distance. It selects nearby vehicles with idle computing resources and processes the tasks in parallel. In vehicular edge computing networks, a task is divided into multiple subtasks before being offloaded to complete the lengthy computation-intensive tasks. Therefore, Qafzez et al. (2022) utilize numerous vehicular edge computing (VEC) network scenarios to test the effectiveness of the suggested approach.

The vehicular fog computing (VFC) architecture, which functions as an interface between mobile devices and fog vehicles, is a three-tiered system designed to assign tasks to the roadside unit (RSU). The mobile devices complete these tasks at the bottom of the hierarchy. With consideration for task deadlines, its goal is to maximize the revenue generated for the automotive industry. Additionally, an orderly network of cars is established and updated frequently to complete the tasks in a time-constrained manner.

This work introduces the Fuzzy Logic Task Offloading Management System (FTOMS); a cloud-MEC collaborative task offloading management system based on fuzzy decision-making that takes advantage of nearby edge servers and robust remote cloud computing capabilities. Choosing the right target node for task offloading while considering latency sensitivity, server capacity, and network condition is the FTOMS scheme’s main goal. In particular, our proposed scheme can dynamically decide which MEC servers to offload high resource-demand, delay-tolerant tasks to a remote cloud server and which to offload delay-sensitive tasks to, like local or nearby servers (Soleymani et al., 2017).

Research contributions

The approach is targeted towards Edge devices, Cloud servers, and local nodes for task offloading decisions based on bandwidth, CPU utilization, task length, and task size. The proposed FI-RBTOM model uses fuzzy logic to determine whether tasks should be offloaded to Edge servers, Cloud servers, or processed locally. A novel Fuzzy Inference Rule-Based Task Offloading Model (FI-RBTOM) is proposed to address the problem of computational task offloading decisions to either compute the task locally or offload the task over an edge or cloud server, considering four main parameters: bandwidth, CPU utilization, task size, and task length. The uniqueness of the proposed model comes from its use of deriving a few fuzzy logic rules against selective key parameters. By formulating and testing these static rules initially, the model predictability of offloading decision is taken. This approach prioritizes efficiency and quality, which are the main goals of this research. FI-RBTOM is designed using a fuzzy inference system (FIS) and a MATLAB (The MathWorks Inc., Natick, NY, USA) simulation tool. A sample dataset of over 100 entries is generated, based on four major parameters, as mentioned in Table 1. Additionally, the dataset is divided into two portions in a ratio of 70:30 for training and testing, to validate the output and reduce the error rate. This approach ensures that the proposed system effectively reduces task computational delays and maintains quality of service.

Table 1 Fuzzy input variables with (fuzzy set and ranges).

Input variable	Notation	Fuzzy set	Ranges	
Bandwidth (Mbps)	α	Low	0–30	
Medium	30–60	
High	60–100	
CPU utilization (%)	β	Low	0–30	
Medium	30–60	
High	60–100	
Task length (GIGA)	λ	Low	0–30	
Medium	30–60	
High	60–100	
Task size (GI)	Ω	Light	0–30	
Normal	30–60	
Heavy	60–100	

Fuzzy logic presents a compelling approach to convey complex operational details in a high-level, human-understandable manner, all based on widely recognized principles. Fuzzy rules are used in our approach to specify the appropriate task offloading activities in terms of networking, computation, and task needs at hand. This allows us to determine the task execution site inside the overall edge computing system.

The following summarizes the key primary contributions:

(1) Designed a novel (FI-RBTOM) model for task offloading.

(2) Training and testing of the proposed (FI-RBTOM) model is done in MATLAB.

Advantages of proposed approach

The fuzzy logic rule-based task offloading approach offers several advantages: 1) Handling uncertainty: Fuzzy logic effectively manages uncertainty and imprecision, making it suitable for environments with variable conditions, such as those in edge computing.

2) Flexibility: The rule-based nature allows for easy adaptation to different scenarios and requirements, enabling customization based on specific application needs.

3) Improved decision-making: By incorporating expert knowledge through fuzzy rules, the approach enhances decision-making processes, leading to more effective task allocation.

4) Real-time processing: Fuzzy logic systems can process data and make decisions in real-time, which is crucial for time-sensitive applications.

5) Reduced complexity: The use of fuzzy rules simplifies complex decision-making processes, making it easier to understand and implement.

6) Enhanced quality of service: By optimizing task offloading based on multiple criteria (e.g., bandwidth, CPU utilization), the approach can improve overall system performance and maintain quality of service.

7) Scalability: The approach can easily scale to accommodate an increasing number of devices and tasks without significant restructuring.

Limitations of the existing study

The FI-RBTOM model proposes a fuzzy rule-based task offloading approach for edge computing. It aims to optimize task allocation between edge, local, and cloud servers based on bandwidth, CPU utilization, task length, and task size. The model achieved an overall error rate of 0.39875 in MATLAB simulations. Future work includes enhancing accuracy with neural fuzzy architectures and applying reinforcement learning for task offloading optimization. The model can be extended for UAVs, Fog, and Edge computing environments. The study addresses challenges in IoT applications and resource management for efficient task distribution.

The authors have tested the initial model in the MATLAB fuzzy logic toolbox and plan to test the rules further by applying a simulation-based environment using CloudSim simulator as highlighted in future directions as well currently due to some academic time bound this could not be completed but it’s in process. Performance prediction is further complicated by the unpredictability of hardware, network conditions, and user requirements. There might be some dynamic factors that could affect usability, i.e., fluctuating network conditions and erratic user behavior.

Literature review

Mobile edge computing (MEC) networks depend on two primary components: task offloading and efficient resource allocation. In accordance with previous research, these can be classified into three main categories based on previous research: partial offloading (where the task is split up into multiple parts simultaneously for offloading or local computing), collaborative task offloading (where the task incorporates with the cloud), and binary or full offloading (wherein the task is unable to split up during processing) (Delowar Hossain et al., 2021).

To decrease latency and the rate at which computing tasks fail to describe, an accurate fuzzy logic decision-based task offloading method has been suggested to enable the use of Internet of Things applications. A mobile edge orchestrator supported by a fuzzy foundation is in charge of managing end-user applications (Nguyen et al., 2020).

To boost the efficiency of managing computational resources, the authors of this article suggested a fuzzy-based mobile edge orchestrator method for flexible computation offloading for Internet of Things applications. By network conditions, computational capacity, and task specifications, a fuzzy-based MEO enables edge workload orchestration actions to decide whether to transfer the mobile user’s tasks to nearby edge, local edge, or cloud servers (Nguyen et al., 2020).

Long and computationally demanding tasks can be completed ahead of schedule by employing the concept of task replication, which is examined by the investigators. This approach selects multiple service vehicles to host the simultaneous processing of a single task, with the final result being returned by a single service vehicle (Li, Tao & Chen, 2020; Chen & Xu, 2019; Sun et al., 2018; Jiang et al., 2018).

Qafzez et al. (2022) tackles the problem of resource management and offer an intelligent architecture that makes use of SDN and FL techniques to manage networking, cloud-fog-edge computing, and storage resources effectively and efficiently in VANETs. The proposed SDN-VANETs approach uses fuzzy logic to manage resources in real time while handling ambiguity and inaccuracy.

This article presents a robust decentralized task-unloading approach that considers several key metrics, including relative velocity, available computing resources, link reliability, mobility, and distance. It selects nearby vehicles with idle computing resources and processes the tasks in parallel. In vehicular edge computing networks, a task is divided into multiple subtasks before being offloaded to complete the lengthy computation-intensive tasks. Several VEC network scenarios are used to test the effectiveness of the suggested approach (Bute et al., 2021).

The vehicular fog computing (VFC) architecture, which functions as an interface between mobile devices and fog vehicles, is a three-tiered system that is designed to assign tasks to the roadside unit (RSU). These are the tasks completed by the mobile devices at the bottom of the hierarchy. With consideration for task deadlines, its goal is to maximize the revenue generated for the automotive industry. Additionally, an orderly network of cars is established and updated frequently to complete the tasks in a time-constrained manner (Soleymani et al., 2017).

This work introduces FTOM; a cloud-MEC collaborative task offloading management system based on fuzzy decision-making that takes advantage of nearby edge servers and robust remote cloud computing capabilities. Choosing the right target node for task offloading while taking latency sensitivity, server capacity, and network condition into account is the main goal of the FTOM scheme. In particular, our proposed scheme can dynamically decide which MEC servers to offload high resource-demand, delay-tolerant tasks to a remote cloud server and which to offload delay-sensitive tasks to, like local or nearby servers (Rai, Vemireddy & Rout, 2021).

Therefore, based on an extensive literature review, a fuzzy logic-based task offloading management system that is trained and tested using Neuro-fuzzy techniques to reduce the error rate in the proposed model has not yet been explored in this domain. To address the task-offloading issue in the edge-computing environment, we propose the Fuzzy Logic Task Offloading Inference System in this study. It is regarded as the most effective method for managing dynamic, unreliable structures due to its versatility in producing precise mathematical models. In addition, the computational complexity of fuzzy logic-based methods is minimal compared to alternative decision-making processes.

Methodology

In this article, the fuzzy logic-based task-offloading algorithm is used to make efficient decisions for offloading the computational task over a suitable server while considering the four major input variables (bandwidth, CPU utilization, task length, and task size). The proposed fuzzy logic inference system, which determines where to offload the computational task with a limited error rate, is used to simulate FI-RBTOM. It allocates distinct thread-hold values to input variables. This technique offloads task data collected from a variety of Internet of Things devices, including end users, using rule-based fuzzy logic.

System model of UAV-based task offloading layer-based architecture

The authors proposed a system model that is based on a tier architecture, which contains three key layers, the UAVS layer, the Edge layer, and the Cloud layer. In UAVs, a layer cluster of UAVs is connected to the Edger layer by using WLAN. On the other hand, the edge layer consists of MEC Servers that communicate with each other by utilizing MAN, which helps with task distribution between the edge server and the cloud. At last, the cloud layer is connected with both layers by a router with WAN to entire layers as illustrated in Fig. 1 below. We have designed three layer-based architectures with reference to processing decisions taken under a fuzzy system i.e., to edge, cloud, and local servers. To meet this, need a number of servers deployed at the top two layers of the architecture. This framework provides UAV clusters in dynamic environments with high mobility and irregular traffic. Task offloading to edge computing often suffers from imbalanced task scheduling and resource allocation, which causes task failure.

Figure 1 Task offloading layer based architecture.

The task offloading model based on fuzzy logic

The investigators of this research study have created and presented a fuzzy logic rule-based task offloading system model (FI-RBTOM) to decide whether the task should be computed at local processing on node, or it should be offloaded over Edge or Cloud server to maximize throughput of the system with minimum drop ratio of the tasks. The proposed model uses four parameters (bandwidth, CPU utilization, task size, and task length) as input to the model with various units as bandwidth is considered in megabytes per second (Mbps), CPU utilization in percentage, task size in Mbps, and task length in generalized instruction process (GIP). It is often used as a unit to represent task length or workload in computing environments, which consists of nine fuzzy rules designed for decision optimization. The primary components of the suggested fuzzy logic task of the model are the singleton fuzzifier, product inference system, centroid defuzzified, and membership functions. As presented in Fig. 2, the FI-RBTOM fuzzy logic task offloading model appropriately generates three options for task offloading based on four input parameters such as bandwidth, CPU utilization, task length, and task size.

Figure 2 Block diagram (FI-RBTOM) model.

Although optimization methods can occasionally be used to obtain membership functions that more closely match data or meet particular performance objectives, they do not always provide complete optimality. The membership functions of fuzzy sets were developed based on empirical data and a literature review tailored. While we strive for optimality, it is context-dependent, and a membership function is generated and gives proof of the validity of the defined thresholds as prescribed under Table 1: fuzzy input variables with (fuzzy set and ranges) to the inherent uncertainty in fuzzy logic. The authors have validated these functions through simulations and found that designed fuzzy logic rules produces less error rate of 0.39875% to meet the accuracy of output decision to offload task.

The fuzzy inference system (FIS) relies on four main variables, as the following equation shows:

A=(μ,π,£,¥)

In this case, µ stands for bandwidth, π for CPU utilization, £ for task length, and ¥ for task size. Fuzzy logic inference systems can leverage a wide range of variables, which are described in Table 1 with fuzzy sets, and ranges. Applying the primary task offloading settings affects the proposed system’s overall performance (Sonmez, Ozgovde & Ersoy, 2019).

Flow chart of task offloading system

The proposed FI-RBTOM system receives task-offloading requests from IoT devices, which are installed at various locations in the form of clusters. After collecting offloading requests from IoTs proposed system offloads the task to desired computing locations based on selected thread hold values. To determine where to forward the task for computing, rule-based logic is applied in the subsequent step. If the bandwidth, CPU utilization, task length, and task size are in between 0 to 30 then offload the computing job to the local server and terminate the execution process. Alternatively, the computing task ought to be dropped to an edge server if the value ranges of both input variables lie between 30 and 60. If not, it ought to be transferred to a local server. In the third step, it is better to execute the computing task to the cloud server if all of the variables’ values fall between 60 and 100; otherwise, it is better to execute it to the edge server and end the simulation there. The process for the suggested FLOMS is shown in Fig. 3.

Figure 3 Flowchart of FI-RBTOM.

Pseudo code for task offloading

  V=Input variables (Bandwidth, CPU Utilization, Task Size, Task Length)	
  Z = Offloading Decision (Local, Edge & Cloud)	
  1) Initialization:	
        Read Bandwidth	
        Read CPU Utilization	
        Read Task Size	
        Read Task Length	
  2) Formula to Calculate Threshold Value against V (Best Case), (Average Case) and (Worst Case)	
        (Best Case):	
              Bandwidth = 30* Total Bandwidth / 100.	
              CPU Utilization = 30 * total CPU Utilization /100.	
               Task Size = 30 * total Task Size /100.	
              Task Length = 30 * total Task Length /100.	
              V = (BW+CU+TS+TL) /100	
              If V== range 0 to 30	
                    Then Z=Local Server	
              Else if	
        (Average Case):	
            Bandwidth = 60 * 100 / 100	
            CPU Utilization = 60 * total CPU Utilization /100.	
            Task Size = 60 * total Task Size /100.	
            Task Length = 60 * total Task Length /100.	
            V= (BW+CU+TS+TL)/100	
            If V== range 30 to 60	
                  Then Z=Edge Server	
            Else	
       (Worst Case):	
          High Bandwidth 100 * 100 / 100.	
          Utilization = 100 * total CPU Utilization /100.	
          Task Size = 100 * total Task Size /100.	
          Task Length = 100 * total Task Length /100	
          V= (BW+CU+TS+TL)/100	
          If V== range 60 to 100	
                Then Z==Cloud Server	
          Else	
          Print (“Task Could No Execute”).	
          End If	
          End;	

Simulation of fi-rbtom in matlab

The proposed FI-RBTOM model is simulated using the MATLAB R2022b x64 version MATLAB (Version 2022) simulator to conduct simulations on a Windows 10 (64-bit) operating system. Both of these software systems are utilized during the simulation process. Additionally, the simulation is performed on a PC with the following configurations: The fuzzy inference system in line with the specified input, output, and membership functions regarding rules as per the methodology section, as illustrated in Fig. 4. Intel(R) Core(TM) m3-7Y30 CPU @ 1.00 GHz (1.61 GHz turbo)

Installed RAM: 8.00 GB.

Figure 4 Fuzzy inference system (FI-RBTOM).

Fuzzification by using MATLAB

Membership functions fuzzies a crisp value, converting it into fuzzy linguistic variables. This study assesses four input variables: bandwidth, CPU utilization, task size, and task length, in addition to one output, the offloading decision. After a decision, rules are put into action. In addition, four inputs (bandwidth, CPU utilization, task size, and task length) that are included in simulation work are used to assess the offloading model designed by FI-RBTOM.

Membership function of bandwidth

The bandwidth is configured as Fig. 5A illustrates the first three linguistic variables: Low, Medium, and High. The value of the change in bandwidth rate from 0 to 100 Mbps is displayed on the x-axis, while the degree of membership function from 0 to 1 is displayed on the y-axis.

Figure 5 Membership functions (FI-RBTOM) for edge computing.

Membership function of CPU utilization

The three linguistic variables (Low, Medium, and High) and their respective CPU utilization initialization are shown in Fig. 5B. The CPU utilization rate change is plotted on the x-axis from 0 to 100, while the degree of membership function is displayed on the y-axis, which ranges from 0 to 1.

Membership function of task size

The task size is started at the point where the x-axis provides the value of change of task size from 0 to 100 and the y-axis shows the degree of membership function from 0 to 1, as can be seen in Fig. 5C represents the three linguistic variables Low, Medium, and High.

Membership function of task length

Figure 5D shows three linguistic variables of task length: Low, Medium, and High. The initialization point for the assigned task length is where the x-axis shows the task length rate change value from 0 to 100 and the y-axis shows the degree of membership function from 0 to 1. The ranges of the linguistic variables are defined in the following manner in the table. There are three ranges: Low (from 0 to 30), Medium (from 30 to 60), and High (from 60 to 100).

Membership function of offloading decision

The three language variables (Local processing, Edge computing, and Cloud computing) are illustrated in Fig. 5E. The degree of membership function, which ranges from 0 to 1, is shown on the y-axis, while the x-axis spans from 0% to 100%, guiding the offloading decisions for computing tasks. When the values are low, between 0% and 30%, the task is offloaded to the local server. If the task falls between 30% and 60%, it is processed by the edge server. When it is in the range of 60% to 100%, the computation occurs on the cloud server.

Results analysis and discussion

Many studies have been successfully conducted to assess the effectiveness of the recommended strategy. To identify a reliable offloading decision based on four important parameters employing the nine rules that were previously addressed in the methodology section.

As shown in Table 2, in Experiment 1, the Bandwidth variable is set to 1, the CPU utilization value is 2, the task size is 3, and the task length is 7. The output of this experiment is 17.2, which suggests that the task should be transferred to the local server. In Experiment 2, the bandwidth is equal to 2, the CPU utilization is set to 1, the task size is 1, and the Task Length is also 1, leading to a recommendation to offload the task to the local server.

Table 2 Experiment results of FI-RBTOM.

Experiment no	Bandwidth	CPU utilization	Task size	Task length	Output	Offload decision	
1	1	2	3	7	17.2	Local server	
2	2	1	1	1	17.2	Local server	
3	3	1	2	3	18.1	Local server	
4	4	5	6	7	19.8	Local server	
5	5	1	2	3	19.8	Local server	
6	6	7	8	9	21.3	Local server	
7	9.8	8.8	7.7	6.6	25.1	Local server	
8	11	10	8	9	23.6	Edge server	
9	15	14	13	12	23.6	Edge server	
10	32	34	36	38	32.1	Edge server	
11	43	43	43	43	37.4	Edge server	
12	50	50	50	50	45	Edge server	
13	55	54	58	59	51	Edge server	
14	60	62	64	68	57.9	Edge server	
15	70	72	74	78	64.4	Cloud server	
16	80	83	86	89	70.8	Cloud server	
17	95	94	93	92	75.5	Cloud server	
18	98	86	75	45	76.2	Cloud server	
19	96	95	94	93	79.3	Cloud server	
20	97	96	95	94	77.2	Cloud server	
21	98	97	96	97	78.1	Cloud server	
22	99	98	97	96	79	Cloud server	
23	100	99	98	97	80	Cloud server	

In Experiment 7, the values for bandwidth, CPU utilization, task size, and task length are all set to 43, indicating that the task should be offloaded to the Edge server. In Experiment 8, the bandwidth is 44, the CPU utilization is 45, the task size is 56, and the task length is 44 here, the output suggests that the task should also be offloaded to the Edge server.

In Experiment 19, the bandwidth value is 99, CPU utilization is 98, and the task size and task length values are 97 and 96, respectively. The output of 79 indicates that both computational tasks should be offloaded to the Cloud server. Lastly, in the final experiment, the bandwidth is set to 100, CPU utilization to 99, and the task size and task length values are 98 and 97, allowing the IoT device to offload the computational task to the Cloud server.

Table 3 represents, how the proposed FI-RBTOM system is implemented in MATLAB against nine rules formulated for four linguistic input variables as mentioned under the methodology section and based on the proposed model the decision to either process job locally or to be offloaded over cloud and edge server has been taken.

Table 3 Fuzzy rules for FI-RBTOM.

Rule no	Bandwidth	Task length	CPU utilization	Task size	Decision	
1	Low	Low	Low	Light	Local processing	
2	Low	Medium	Medium	Normal	Edge server	
3	Low	High	High	Heavy	Edge server	
4	Medium	Low	Low	Light	Edge server	
5	Medium	Medium	Medium	Normal	Edge server	
6	Medium	High	High	Heavy	Cloud server	
7	High	Low	Low	Light	Edge server	
8	High	Medium	Medium	Normal	Cloud server	
9	High	High	High	Heavy	Cloud server	

As displayed in Fig. 6, the outcome of experiment 11 is displayed in the rule viewer where the column represents the variables, and the row represents the rule. The first column of the rule viewer denotes the Bandwidth, the second column represents the CPU utilization, and the third and fourth columns express the task length and size, accordingly. These four columns represented input variables with the same values. The fifth column shows output, which represents the better offloading decision based on the value of input variables. According to the figure, the value of bandwidth is 50 Mbps, CPU utilization is 50, task length is 50 and task size is 50. The output of this experiment is 45%, which represents the range of edge computing. Therefore, the offloading task is assigned to the edge server for further computation.

Figure 6 Task offloading in experiment 1 for FI-RBTOM.

As demonstrated in Fig. 7, in Experiment 7, the variables are shown in columns while the rules are presented in rows. The first column of the rule viewer denotes the bandwidth, the second column represents the CPU utilization, and third column represents the task length, at last, the fourth column represents the task size. These four columns represented input variables with the same values. The fifth column shows output, which represents the better offloading decision based on the value of input variables. According to the figure, the value of bandwidth is 9.8 Mbps, CPU utilization is 8.8, task length is 7.7 and task size is 6.6. The output of this experiment is 25.1%, which represents the range of local computing. Therefore, the offloading task is assigned to the local server for further computation.

Figure 7 Offloading decision in experiment 2 (FI-RBTOM).

As presented in Fig. 8 the outcome of experiment 18 is displayed in the rule viewer. Where the column represents the variables and the row represents the rule. The first column of the rule viewer denotes the Bandwidth, the second column represents the CPU utilization, and the third column denotes the task length, finally, the task size is presented in the fourth column. These four columns represented input variables with the same values. The fifth column shows output, which represents the better offloading decision based on the value of input variables. According to the figure, the value of bandwidth is 98 Mbps, CPU utilization is 86, task length is 75 and task size is 45. The output of this experiment is 76.2%, which represents the range of cloud computing. Therefore, the offloading task is assigned to the cloud server for further computation.

Figure 8 Offloading decision in experiment 3 (FI-RBTOM).

The FIS output data appears above the testing data, as can be seen. With an average testing error rate of 0.39875, as shown in Fig. 9 below, it indicates that our FI-RBTOM is adequately trained. The data is represented by blue, which is the actual data; additionally, red stars, which correspond to the anticipated data, appear correctly over the actual data. This indicates that our system is correctly completing the task with a letter bit testing error in the system. The development of a system that achieves 100% accuracy is a very challenging task.

Figure 9 Average testing result of FI-RBTOM.

Conclusions and future work

The sophisticated functions of Internet of Things devices are anticipated to be substituted by edge computing and have restricted processing, storage, and battery capacity. Mobile devices, edge servers, and the dynamic nature of the network all negatively impact system performance. The researchers suggest several works offloading strategies to address these issues, managing edge resources to improve performance. The task offloading problem cannot be resolved by offline optimization techniques since in such a dynamic environment, the demand of mobile devices is unpredictable. This makes resource management and task offloading challenging online issues that need to be resolved. In this study, research has been done on a fuzzy logic inference system based on four main parameters that are trained and verified over neural fuzzy architecture. Using WLAN access technologies and fuzzy logic-based inference systems, which precisely determine which servers to conduct operations on; IoT devices delegate their workloads to edge or cloud servers. Writers used MATLAB to put up a simulated environment in which assessed our job offloading inference system’s performance utilizing fuzzy logic. The simulation results obtained from the FI-RBTOM achieved a 0.39675 error rate concerning 75% training and 25% testing.

Future directions

In the future, the accuracy of the FI-RBTOM could be further optimized by proposing a neural fuzzy-based architecture to gather data samples from the proposed fuzzy logic inference system in this research study. The authors have further planned to improve this work using an adaptive neuro-fuzzy inference system with applying reinforcement learning techniques to optimize the task-offloading problem in the Clouds simulator for comparison with existing similar studies done earlier. Analyzing the effects of task migration between local, edge, and cloud servers might also be interesting. This (FI-RBTOM) model can be used in the future for efficient task offloading decisions under UAVs (Unmanned Aerial Vehicles), Fog, and Edge computing environments for local, edge, and cloud server utilization for the distribution of tasks for efficient completion.

Supplemental Information

Supplemental Information 1 The execution file.

Supplemental Information 2 The sample dataset with 100 Values on Bandwidth, CPU Utilization, TS, TL and Output.

The first author is thankful to the leadership support of other authors in this research.

Additional Information and Declarations

Competing Interests

The authors declare that they have no competing interests.

Author Contributions

Kashif Ibrahim conceived and designed the experiments, analyzed the data, performed the computation work, authored or reviewed drafts of the article, and approved the final draft.

Ahthasham Sajid performed the experiments, performed the computation work, authored or reviewed drafts of the article, and approved the final draft.

Ihsan Ullah performed the experiments, performed the computation work, prepared figures and/or tables, and approved the final draft.

Inam Ullah Khan analyzed the data, performed the computation work, prepared figures and/or tables, authored or reviewed drafts of the article, and approved the final draft.

Keshav Kaushik conceived and designed the experiments, performed the computation work, authored or reviewed drafts of the article, and approved the final draft.

S. S. Askar performed the computation work, authored or reviewed drafts of the article, and approved the final draft.

Mohamed Abouhawwash performed the computation work, authored or reviewed drafts of the article, and approved the final draft.

Data Availability

The following information was supplied regarding data availability:

The raw measurements are available in the Supplemental Files.

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
