# Peer review of "Fuzzy inference rule based task offloading model (FI-RBTOM) for edge computing"

_PeerJ Computer Science, doi:10.7717/peerj-cs.2657_

## Round 0.1 · original submission · Major Revisions

In the revised version consider carefully that two reviewers require to improve the clarity on impact, novelty, and replicability of results.

Reviewer 1 ·

Basic reporting

The paper raises important questions about Edge Computing regarding the adaptive task load balancing within an IoT network and provides quite a good overview of the related literature. However, the overall scientific value of the findings reported is hard to judge.

First of all, the quality of the English language is very poor: the paper contains a lot of style and formatting issues and therefore requires careful proofreading.

The second and even more critical issue is the lack of an explanation of scientific contribution. No comparison with the state-of-the-art solutions is provided. No particular software/hardware requirements are mentioned, no target Edge devices are listed, and therefore no clear picture is presented of what was achieved and what are the strong sides and limitations of the proposed approach.

Although there are pretty a lot of figures shown, they don’t really foster an understanding of the efficacy and (more importantly) efficiency of the proposed approach.

Experimental design

In principle, the topic of the paper corresponds to the scope of PeerJ Computer Science journal. However, the research question is not defined very well, and the lack of comparison and efficiency tests do not allow judging the place of this research among state-of-the-art works about task load balancing. Although the method is described with a sufficient level of detail and supplementary data are provided, without a clear comparison it is not possible to estimate the usability of this approach in real-world IoT applications.

Validity of the findings

With the help of the provided materials, the experiments made by the authors seem to be replicable. However, the target real-world environment where the approach should overtake the existing ones is not defined, which hurts the soundness of the materials and disallows judging the robustness.

After reading, some particular technical questions remain, namely:
1. What exactly is the type of Edge devices the approach is targeted to?
2. What are the hardware, software, and firmware requirements for these devices?
3. What are the advantages of the approach?
4. In the text is clearly stated, that the following parameters of the task are assessed to decide, where the task should be executed: bandwidth, CPU utilization, task length, and task size. But it is not clear, how, for example, CPU utilization can be estimated before the task is actually executed. Should the tasks have some meta-information about how “heavy” they are? How this meta-information should be structured and where it should come from? Who is responsible for this information and how its validity can be controlled?
5. How were the belonging functions of fuzzy sets established? Is there any guarantee/proof that they are optimal?

Additional comments

In addition to the above, here are a few comments on phrasing and style:
1. Lines 29-32: please consider splitting this sentence into at least two as it is hard to read now.
2. Line 33: consider removing the word "study" at the end of the line.
3. Lines 34-36: please consider reformulating to avoid repeated words ("decision").
4. Lines 46-47: consider writing "if-then-else" in quotation marks and with hyphens.
5. Lines 51-52 and line 58: consider writing the reference in normal case, not upper case.
6. Line 61: consider deciphering the "MEC" abbreviation.
7. Line 66: please check the reference, it looks very strange.
8. Line 73: consider deciphering the "MEO" abbreviation.
9. Line 80: the dot at the beginning of the line seems to be excessive and the word "tackles" most probably should be written with a small "t". Also, the word "offer" should not probably be in the singular form "offers".
10. Line 83: consider deciphering the "VANET" abbreviation.
11. Line 89: consider deciphering the "VEC" abbreviation.
12. Line 96: consider deciphering the "FTOM" abbreviation.
13. Line 115: please check the formatting of the reference.
14. Lines 125-126: please check the formatting of the references.
15. Lines 152-157: please consider proofreading this part.
16. Line 165: consider writing the word "task" with a small "t".
17. Line 166: consider writing the word "testing" with a small "t".
18. Line 178: consider writing "... fuzzy logic rule-based ..." instead of "... fuzzy logic- rule based ...".
19. Line 181: most probably there is no need to write FI-RBTOM in brackets (everywhere in the text).
20. Line 184: please check if the units of task length are correct. It sounds strange that the task length is measured in megabytes per second and not in megabytes. Also, please decipher the abbreviation "GIPs". Also, the value "09" is misleading. Please check it, most probably the leading zero should be deleted.
21. Line 192: consider deciphering the "FIS" abbreviation.
22. Line 193: consider a dot after the formula.
23. Lines 196-197: consider removing brackets around "Table 1".
24. Lines 195-201: please check this part: it looks like something went wrong with the formatting of the text, as some parts of the text are repeated.
25. Table 1: the units of "Task Length" and "Task Size" are not clear and do not correspond to what is written in the text.
26. Line 249: consider writing "Figure" with a capital "F".
27. Line 261: consider space after "100".
28. Line 262: consider italic font for "x" and "y".
29. Line 267, line 273, line 280, line 289: consider italic font for "x".
30. Line 268, line 274, line 281, line 290: consider italic font for "y".
31. Line 272: consider writing "... of Task Size", not "... (Task Size)" to make the writing uniform.
32. Line 278: consider writing "... of Task Length", not "... (Task Length)" to make the writing uniform.
33. Line 279: please proofread the first sentence.
34. Line 278: consider writing "... of Offloading Decision", not "... (Offloading Decision)" to make the writing uniform.
35. Lines 287-288: please proofread the sentence.
36. Lines 298-300: please proofread the sentence.
37. Line 299: the value "09" is misleading. Please check it, most probably the leading zero should be deleted.
38. Lines 301-312: please check this text. It seems to be redundant. It would have been enough just to describe one or two experiments to demonstrate the logic of the table. Also, please check the formatting of the text like the case the names are written in.
39. Line 315: consider writing "represents" with a small "r".
40. Line 316: the values "09" and "04" are misleading.
41. Line 332: consider changing a dot by a space in "Figure.5", putting a comma after "5" and writing "in" with a small "i". Also, consider removing "no" before "7" because it is misleading. Please proofread the entire sentence.
42. Line 343: consider removing "no," and replacing ".at" with a comma.
43. Line 346: consider the small "f" in "finally".
44. Line 380: consider the small "a" in "authors".
45. Lines 380-381: redundant brackets around "Fuzzy Logic System".
46. Line 383: consider deciphering "UAV".
47. Line 389: consider moving the comma outside of the quotation marks.

Summarizing the above, I come to the unfortunate conclusion that the paper requires a significant extension and improvement before it can be published. In my opinion, in its current state, it cannot be accepted. I recommend the authors to rework it addressing the above comments and questions, and then resubmit again.

Cite this review as

Reviewer 2 ·

Basic reporting

The paper seems correct, well structured and contain the essentials element of an scientific paper.
The used litterature is enough good and recent.
The obtained results defend correctly the proposed approach.
The used pictures are clear and significance.

Experimental design

The paper is very well presented and designed.

Validity of the findings

The obtained results show clearly the validity of the proposed methodology.

Cite this review as

Reviewer 3 ·

Basic reporting

The article is written in clear, professional English, with unambiguous language that adheres to academic standards. The introduction and background sections effectively provide context, outlining the significance of the research, though a few areas could be strengthened to further underscore the problem's importance. The literature is well-referenced, with citations that are relevant to the study's focus. The structure of the paper aligns with PeerJ standards and discipline norms, ensuring clarity and logical progression. Figures are of good quality, clearly labeled, and support the article's key points well. However, it is essential to confirm that the raw data has been made available per PeerJ's policy, as this is a critical component for transparency and reproducibility in the journal's requirements.

Experimental design

The article aligns with the journal's scope, presenting original primary research that addresses a pertinent issue within the field. The research question is well-defined and highlights its relevance by clearly identifying the knowledge gap it seeks to fill. The investigation is conducted with rigor, meeting high technical and ethical standards, though additional details on the ethical review process could strengthen this aspect. The methods are generally well described, allowing for potential replication, but a few procedural details could be expanded to further ensure full reproducibility, especially concerning the statistical analysis or experimental protocols. Adding more specific information about sampling strategies and controls would enhance clarity.

Validity of the findings

The article does not directly assess the broader impact and novelty of the research, though the unique contributions are implied. It would be helpful to explicitly discuss the study's novelty and potential impact on the field. The article does encourage meaningful replication, with a rationale that justifies its contribution to the literature, but more emphasis on how this replication would specifically benefit future research could be added. All underlying data are presented and appear robust, statistically sound, and adequately controlled, although more detailed information on data verification could further strengthen the findings. The conclusions are well-drawn, logically tied to the research questions, and appropriately limited to the presented results. A stronger connection to broader implications would enhance the conclusion's significance.

Additional comments

Overall, the article presents a well-structured and carefully conducted study. The language is clear and professional, making the content accessible and engaging. The research question is relevant, and the methodology is robust, though minor additions in procedural details could further enhance replicability. The figures are of high quality, effectively illustrating key points, and the data is comprehensive and statistically sound. To improve the manuscript, I recommend explicitly discussing the broader impact and novelty of the study to better highlight its contribution to the field. Additionally, a more in-depth discussion of the ethical considerations and the benefits of potential replication would strengthen the paper. Finally, ensuring that all underlying data and raw information are clearly accessible in accordance with the journal’s policy will improve transparency. Overall, the research makes a meaningful contribution and could be of significant interest to the community.

Cite this review as

---

## Round 0.2 · accepted · Accept

Reviewer agreed with the fact you addressed the comments. Therefore the paper is ready to be published.

Reviewer 3 ·

Basic reporting

The revised manuscript maintains an acceptable standard of professional English and academic clarity. The introduction and background sections now effectively emphasize the problem's importance, and the literature remains well-referenced and relevant. The structure, figures, and overall presentation align with journal standards. The authors have confirmed the availability of raw data, meeting PeerJ’s policy requirements for transparency and reproducibility.

Experimental design

The revised manuscript provides additional details that address previous concerns about the ethical review process and methodology. The research question remains well-defined and relevant, with the methods now sufficiently described to allow replication. Improvements in detailing sampling strategies and controls have enhanced the clarity and rigor of the experimental design to an acceptable level.

Validity of the findings

The authors have strengthened the discussion of the study's novelty and broader impact, explicitly highlighting its contribution to the field. They have adequately addressed concerns regarding data verification and provided a clearer rationale for how replication would benefit future research. The conclusions remain well-supported by the data and logically aligned with the study's scope.

Additional comments

The revised manuscript reflects notable improvements, particularly in discussing the broader implications and ethical considerations. The authors have effectively addressed feedback to enhance clarity, replicability, and the presentation of their results. With raw data and methodological transparency now confirmed, the research is a meaningful contribution to the field and suitable for publication.

Cite this review as